

# Primary and secondary sources of ambient formaldehyde in the Yangtze River Delta based on OMPS observation

Wenjing Su[1], Cheng Liu[1, 2, 3, 4], Qihou Hu[2], Shaohua Zhao[5], Youwen Sun[2], Wei Wang[2], Yizhi Zhu[2], Jianguo Liu[2, 3], Jhoon Kim[6]

[1]School of Earth and Space Sciences, University of Science and Technology of China, Hefei, 230026, China
[2]Key Lab of Environmental Optics & Technology, Anhui Institute of Optics and Fine Mechanics, Chinese Academy of Sciences, Hefei, 230031, China
[3]Center for Excellence in Regional Atmospheric Environment, Institute of Urban Environment, Chinese Academy of Sciences, Xiamen, 361021, China
[4]Anhui Province Key Laboratory of Polar Environment and Global Change, USTC, Hefei, 230026, China
[5]Satellite Environment Center, State Environmental Protection Key Laboratory of Satellite Remote Sensing, Ministry of Ecology and Environment, Beijing, 100094, China
[6]Department of Atmospheric Sciences, Yonsei University, Seoul, 03722, Korea

*Correspondence to*: Cheng Liu (chliu81@ustc.edu.cn), Qihou Hu (qhhu@aiofm.ac.cn)

**Abstract.** Formaldehyde (HCHO) in the ambient air not only causes cancer but also is an ideal indicator for volatile organic compounds (VOCs) which are major precursors of ozone ($O_3$) and secondary organic aerosol (SOA) near the surface. It is meaningful to differentiate between direct emission and secondary formation for HCHO pollution control and sensitivity study of $O_3$ production. However, the understanding of the sources of HCHO is still poor in China, due to limited measurements of HCHO in the field, both spatially and temporally. In this study, tropospheric HCHO vertical column densities (VCDs) in the Yangtze River Delta (YRD), East China where HCHO pollution is serious were retrieved from Ozone Mapping and Profiler Suite (OMPS) onboard the Suomi National Polar-orbiting Partnership (Suomi-NPP) satellite from 2014 to 2017, and kept good agreement with the tropospheric HCHO columns measured by ground-based high resolution Fourier transform infrared spectrometry (FTS) with the correlation coefficient (R) of 0.78. Based on this, the cancer risk was estimated nationwide and in the YRD region. At least, 7840 people in the YRD region would develop cancer in their lives due to outdoor HCHO exposure, which occupied 23.4 % of total national cancer risk. Besides, the contributions of primary and secondary sources were apportioned, combining with primary and secondary tracers from surface observation. Overall, HCHO from secondary formation contributed most to ambient HCHO and can be regarded as the indication of the VOCs reactivity in Hangzhou and urban areas of Nanjing and Shanghai from 2015 to 2017, due to strong correlation between total HCHO and secondary HCHO. At industrial sites in Nanjing, primary emission influenced most to ambient HCHO in 2015 and showed an obvious decreasing trend. Seasonally, HCHO from secondary formation reached the maximum in summer and minimum in winter. In the spring, summer, and autumn, secondary formation played a curial effect on variation of ambient HCHO in urban regions of Nanjing, Hangzhou, and Shanghai; while in the winter the contribution from secondary formation became less significant. The understanding of the variation of the primary and secondary contributions to ambient HCHO is in favor for a better



understanding the role of HCHO in atmospheric chemistry and formulating effective control measures to decrease HCHO pollution and cancer risk.

## 1 Introduction

In recent years, air pollution has become increasing serious in China (Tang et al., 2012;Shi et al., 2014;Rohde and Muller, 2015) and caused grievous risks for human health (Wang and Mauzerall, 2006). Previous studies indicated that air pollution killed more people worldwide than tuberculosis, AIDS, breast cancer and malaria (Yang et al., 2013; World Health Organization, 2014a, b). According to the U.S. Environmental Protection Agency (EPA), HCHO is identified as the most major carcinogen among the 187 hazardous air pollutants (HAPs), and exposure to as low as 1 $\mu gm^{-3}$ (about 0.7 ppb at STP) of HCHO in one's life will cause up to 13-in-a-million lung and nasopharyngeal cancer risk (Zhu et al., 2017a). HCHO not only affects public health directly but also plays a significant role in atmospheric photochemistry (Levy, 1971). Photolysis of HCHO leads to the producing of $HO_x$ radical ($OH+HO_2$) and affects the oxidative capacity of the atmosphere (Volkamer et al., 2010). Besides, HCHO may contribute to causing $O_3$ pollution through the photochemical reaction (Haagen-Smit, 1950;Carter, 1994;Russell et al., 1995) and favor to the formation of secondary organic aerosol (SOA) through providing OH radical (Yang et al., 2018). So monitoring and controlling ambient HCHO concentration are of great significance for public health.

HCHO can be emitted into the atmosphere directly from biogenic sources, such as biomass burning and vegetation (Lee et al., 1997;Finlayson-Pitts and Pitts Jr, 1999;Holzinger et al., 1999;Andreae and Merlet, 2001) and anthropogenic activities, such as vehicles emissions, industrial emissions and coal combustion (Carlier et al., 1986;Williams et al., 1990;Hoekman, 1992;Carter, 1995;Anderson et al., 1996;Kean et al., 2001;Klimont et al., 2002;Reyes et al., 2006;Wei et al., 2008;Wang et al., 2013;Dong et al., 2014;Liu et al., 2017). Direct emission of HCHO is mainly from the incomplete combustion process and is closely related to the emission of CO. Besides, HCHO can also be formed from the atmospheric oxidation of volatile organic compounds (VOCs) (Altshuller, 1993;Carter, 1995;Seinfeld and Pandis, 2016) which leads to the formation of $O_3$ in the meantime (Rasool, 1970;Crutzen, 1979;Warneke et al., 2004;Duan et al., 2008). In order to control HCHO concentrations effectively and improve air quality, it is necessary to determine the primary emission and secondary formation of ambient HCHO.

In previous studies, the ratio of the Ozone Monitoring Instrument (OMI) tropospheric column of HCHO and $NO_2$ was used as the indicator of sensitivity regime of $O_3$ formation (Duncan et al., 2010;Witte et al., 2011;Liu et al., 2016;Su et al., 2017), on the assumption that HCHO is regarded as the proxy for the reactivity of VOCs. Secondary HCHO is produced with the formation of peroxy radicals ($RO_2$), and thus positively correlated with $RO_2$ and can be identified as the proxy for total VOC reactivity (Carter, 1994). So, strictly speaking, only HCHO concentration that is from secondary formation can be used to analyze the sensitivity regime of $O_3$ production, combined with the $NO_2$ concentration. As secondary formation is the dominate source of HCHO on a global scale (Smedt et al., 2008), total HCHO was usually adopted to replace secondary HCHO as the indicator of total VOCs reactivity. However, in some regions with strong human activities, the contribution of primary



emission to ambient HCHO cannot be ignored (Ma et al., 2016). Separation of secondary HCHO from primary HCHO will be in favor for a better understanding of ozone formation sensitivity, especially in urban areas.

Previous studies attempted to identify contributions to ambient HCHO from direct emission and secondary formation through the linear multiple regression analysis. Because of the strong relation of primary HCHO and CO emission and formation of HCHO and $O_3$ from the oxidation of VOCs, CO and $O_3$ can be used as the tracer for primary emission and secondary formation of HCHO, respectively. There were several studies on primary and secondary sources of HCHO using the CO-$O_3$ tracer pair. For instance, contributions from primary and secondary sources to ambient HCHO were characterized by statistical analogy to CO and $O_3$ in Houston, TX in the summer of 2000, in Beijing during the 2008 Olympic Games, and in Hong Kong from winter of 2012 to autumn of 2013 (Friedfeld et al., 2002;Li et al., 2010;Lui et al., 2017). Hong et al. (2018) estimated contributions from primary emission and secondary formation to HCHO measured by MAX-DOAS using CO-$O_x$ ($O_3$+$NO_2$) tracer pair which were measured by Sensor Networks for Air Quality (SNAQ) in the winter of 2015 in Yangtze River Delta. Garcia et al. (2006) indicated that using CO-CHOCHO tracer pair to separate different sources of HCHO is more reasonable than that using CO-$O_3$ because CHOCHO has the similar lifetime in the atmosphere to HCHO. However, it is not accessible to continuous observation of CHOCHO.

Previous studies often concentrated on contributions from different sources just at one site and within a short time, due to the constraints from the meantime measurements of HCHO and its tracers for primary emission and secondary formation. In China, CO and $O_3$ can be obtained from the China National Environmental Monitoring Center (CNEMC) Network. However, HCHO is not measured at these stations. There are various analytical methods to measured HCHO concentration, such as Fourier transform infrared spectroscopy (FTIR) (Lawson et al., 1990), differential optical absorption spectroscopy (DOAS) (Platt and Perner, 1980), tunable diode laser absorption spectroscopy (TDLAS), 2,4-dinitrophenylhydrazine (DNPH) method which HCHO was captured by DNPH and then analyzed by high-performance liquid chromatograph (HPLC) (Fung and Grosjean, 1981), and proton transfer reaction mass spectrometry (PTR-MS) (Karl et al., 2003). Nevertheless, it is difficult to perform long-term observations of HCHO concentration in a wide range using ground-based measurements. Long-term information of ambient HCHO on a global scale can be obtained accessibly from space-based remote sensor like Global Ozone Monitoring Experiment (GOME) (Martin et al., 2004), Scanning Imaging Absorption Spectrometer for Atmospheric Chartography (SCIAMACHY) (Bovensmann et al., 1999), GOME-2 instruments (De Smedt et al., 2012;De Smedt et al., 2015), Ozone Monitoring Instrument (OMI) (Abad et al., 2015), Ozone Mapping and Profiler Suite (OMPS) (Li et al., 2015a;González Abad et al., 2016) and TROPOspheric Monitoring Instrument (TROPOMI) (De Smedt et al., 2018). While HCHO observations from SCIAMACHY and TROPOMI are only available until 2011 (Shah et al., 2018) and after Oct 2017, respectively. GOME-2A instrument suffers degradation issues (De Smedt et al., 2012) and OMI observations are easily affected by the instrumental "row anomaly" (Abad et al., 2015). And the spatial (80×40 $km^2$) and temporal (within 1.5 days) resolution of GOME-2B is even lower than OMPS (Munro et al., 2016). Overall considering for long-term record and data quality, HCHO observation from OMPS was adopted in this study.

In this paper, we focus on contributions from primary and secondary sources to ambient HCHO from 2015 to 2017 in



Nanjing (capital of Jiangsu province), Hangzhou (capital of Zhejiang province) and Shanghai, which are megacities and have population more than 8 million in the YRD region. The YRD region is located in the alluvial plains where the Yangtze River drains into the East China Sea, consisting Jiangsu and Zhejiang provinces and Shanghai municipality. The YRD region is one of the most important economic and a rapidly developing industry region in China. Along with the very rapid development of

economy and industry, the YRD region is suffering from serious air pollution (Li et al., 2015b;Gao et al., 2016;Wang et al., 2017a;Wang et al., 2018). Details of the measurements are described in Sect. 2. Spatiotemporal distribution of HCHO VCDs in the YRD region, as well as contributions from primary and secondary sources to ambient HCHO are shown in Sect. 3.

## 2 Measurements and methodology

### 2.1 OMPS HCHO observation

Ozone Mapping and Profiling Suite Nadir Mapper (OMPS-NM) which was one of the OMPS suite of instruments was launched on 28 October 2011 onboard the Suomi National Polar-orbiting Partnership (Suomi-NPP) satellite. Suomi-NPP was a polar Sun-synchronous satellite with an average altitude of 824 km (Flynn et al., 2014). It crosses the equator each afternoon at about 13:30 local time (LT) on the ascending node. The OMPS-NM combines a single grating and a 340×740 pixel charge-coupled device (CCD) detector to measure UV radiation every 0.42 nm from 300 to 380 nm with 1.0nm full width at half

maximum (FWHM) resolution. A recent analysis shows that the OMPS-NM sensor has a signal-to-noise ratio of 2000:1 or better at the wavelength range from 320 to 370 nm (Seftor et al., 2014). It has a 110° cross-track field of view (FOV) providing high temporal (daily global coverage) resolution and measurements were combined into 35 cross-track bins giving spatial resolution of 50 ×50 km$^2$ in OMPS-NM standard Earth science mode. OMPS-NPP Nadir Mapper Earth View Level 1B data available at https://ozoneaq.gsfc.nasa.gov/data/omps/ was downloaded to retrieve HCHO Slant Column Density (SCD).

Details of fitting settings for HCHO SCD retrieval was studied in (González Abad et al., 2016). In this study, the vertical profiles of HCHO from WRF-Chem modelling were used to calculate Air Mass Factor (AMF). The configuration of the modelling was described in detail in our previous study (Su et al., 2017). Finally, HCHO VCD is calculated using AMF as follows:

$$VCD=\frac{SCD}{AMF}$$    (1)

### 2.2 FTS tropospheric HCHO column measurement

The ground-based FTS instrument, located in the west of Hefei (117.17° E, 31.9° N), is a candidate site for Network for Detection of Atmospheric Composition Change (NDACC, http://www.ndacc.org/). The global NDACC can perform simultaneous retrieval of mixing ratio profiles for trace gases, such as $O_3$, CO, $CH_4$, $NO_2$, and HCHO, using FTS (Kurylo, 1991;Notholt et al., 1995;Vigouroux et al., 2009), and the retrieval results were widely used for atmospheric chemistry study





and the validation for satellite observation (Yuan et al., 2015;Sun et al., 2017;Wang et al., 2017c;Tian et al., 2018;Sun et al., 2018). The observation system consists of a high resolution Fourier transform spectrometer (IFS125HR), a solar tracker (Tracker-A solar 547), and a weather station (ZENO-3200). The FTS solar spectra measurements are performed over a broad spectral range of 600-4500 cm⁻¹ with a spectral resolution of 0.005 cm⁻¹, and the HCHO spectra is recorded in the range of 2400-3310 cm⁻¹. The vertical profile of HCHO was retrieved using SFIT4 algorithm (version 0.9.4.4). Detailed retrieval settings for HCHO are listed in Sun et al. (2018). Daily priori profiles of pressure, temperature and $H_2O$ are obtained from the National Centers for Environmental Protection and National Center for Atmospheric Research (NCEP/NCAR) reanalysis. And priori profiles of HCHO and its interfering gases have been taken from a dedicated Whole Atmosphere Community Climate Model (WACCM). For HCHO observation, the four micro windows (MW) which are centered at around 2770 cm⁻¹ are selected for satellite validation. And a de-weighting signal to noise ratio (SNR) of 600 was used. The instrument line shape (ILS) is described by analyzing HBr cell spectra using LINEFIT14.5 software. Finally the tropospheric HCHO partial column was calculated by integrating the retrieved profiles with the air-mass profile within 15 km as Eq. (2).

$$PC_{trop} = \int_0^{15} X_r * A_m \tag{2}$$

Where $X_r$ is the retrieved profile of HCHO; $A_m$ is the air-mass profile; and $PC_{trop}$ is the tropospheric HCHO Column. The FTS total systematic error of the HCHO retrieval is less than 3.3 % and total random error is less than 9.6 (Sun et al., 2018). Due to the high accuracy, the trace gases concentrations measured by FTS were widely used to validate corresponding satellite products (Yamamori et al., 2006;Jones et al., 2009;Yurganov et al., 2010;Reuter et al., 2011). Here we used the FTS HCHO measurement to validate OMPS HCHO product.

**2.3 The China National Environmental Monitoring Center (CNEMC) Network**

Surface air pollutants monitored by CNEMC was provided by Ministry of Environment Protection of the People's Republic of China hourly (http://106.37.208.233:20035/). The national air quality monitoring network consisted of about 950 sites in 2013, extending to 1597 sites in 454 major cities in 2017. Three trace gas, including CO, $O_3$, and $SO_2$, inhalable particles ($PM_{10}$), and fine particulate matter ($PM_{2.5}$) are measured simultaneously by all sites which have been used widely used (Wang et al., 2017b;Li et al., 2018;Liu et al., 2018;Lu et al., 2018b). In this study, CO and $O_3$ data from 2104 to 2017 was used. Data quality controls similar to previous studies (Barrero et al., 2015;He et al., 2017;Li et al., 2018) have been applied to remove data outliers. Firstly, all hourly data at a specific monitoring site were transformed into z scores, and then the transformed data ($Z_i$) were removed if they meet one of the condition (1) the absolute $Z_i$ larger than 4 ($|Z_i|>4$) (2) the increment of $Z_i$ from the previous hourly value larger than 9 ($|Z_i-Z_{i-1}|>9$) and (3) the ratio of the z score to its centered moving average of order 3 larger than 2 ($\frac{3\,Z_i}{Z_{i-1}+Z_i+Z_{i+1}}>2$).





## 3 Results

### 3.1 Comparison with FTS tropospheric HCHO

In order to compare the OMPS observation with FTS measurement, the HCHO tropospheric column measured by FTS was averaged around the OMPS satellite overpass time. The OMPS tropospheric HCHO column selected for satellite pixels

within 50 km around the Hefei site and with cloud fraction less than 20 % were averaged to minimize the random noise (Wang et al., 2017d). Fig. 1a shows that tropospheric HCHO VCDs observed by OMPS and FTS are in good agreement (R= 0.78). The coincident time series of both data indicate that OMPS HCHO observation successfully captures maximum HCHO concentration in summer and minimum HCHO concentration in winter (Fig. 1b). However, it is important to note that OMPS HCHO observation underestimates tropospheric HCHO column by 0 % to up 60 % compared with FTS results, especially in

summer, which is consistent with the previous study (De Smedt et al., 2015). This underestimation by satellite may be caused by errors in spectral fitting and AMF calculation (Zhu et al., 2016).

Most of the atmosphere HCHO was concentrated in the troposphere. The monthly averaged vertical profiles of HCHO showed in Fig. 2a indicate that the structure of the HCHO vertical profiles in different months was consistent. HCHO concentration decreased by 70 % with an increase of the height from 0.7 to 2 km and continued to decrease slowly in the

troposphere. The ratio of HCHO column below 1 km in tropospheric HCHO column keeps at ~67 % without seasonal variation (Fig. 2b). HCHO column below 1 km measured by FTS and OMPS tropospheric HCHO column also kept good agreement (R=0.78, Fig. S1 in the Supplementary Information). And the slope of 0.83 for the linear regression between each other was higher than the slope of 0.51 for the regression between OMPS data and the tropospheric HCHO VCD from FTS indicating that OMPS tropospheric HCHO column could represent HCHO column below 1km better. Besides, the mean of the relative

differences between HCHO column below 1 km measured by FTS and Tropospheric HCHO column measured by OMPS is zero. In previous studies, a uniform and constant correction factor was applied to account for the bias between the tropospheric HCHO column observed by satellite and that measured by aircraft (Anderson et al., 2016). In this study, we regard tropospheric HCHO column observed by OMPS as that below 1 km measured by FTS in terms of their numerical value. Assuming that HCHO mixes well within 1 km height, the mixing ratio of HCHO can be calculated as:

$$M(ppbv) = \frac{1.25 \times VCD(\text{molecules cm}^{-2})}{B \times \Delta p(\text{atm})} \qquad (3)$$

where M is the mixing ratio of HCHO; B is unit conversion factor of DU to $molecules\ cm^{-2}$ with the value of $2.688 \times 10^{16}$; VCD is OMPS tropospheric HCHO VCDs; $\Delta p$ is the pressure difference between surface and 1km height. (Ziemke et al., 2001;Lee et al., 2008).

### 3.2 Spatiotemporal distribution of HCHO VCDs throughout The Yangtze River Delta

The average HCHO VCD over China in August, 2017 was $6.50 \times 10^{15}$ molecules cm$^{-2}$ which would lead up to 50 in a





million to develop cancer. Total national cancer risk was estimated through combining averaged HCHO VCDs with population and at least 33500 people in China would develop lung and nasopharyngeal cancer in their lives due to outdoor HCHO exposure. Considering the life expectancy of 76.4 years (2015) in China (http://data.stats.gov.cn/easyquery.htm?cn=C01&zb=A0304&sj=2016, in Chinese), 439 cancer cases per year in China are

caused by high outdoor HCHO concentration. HCHO VCDs in most of western China, e.g. Yunnan province, Tibet, Gansu Province, Qinghai province, Xinjiang Uyghur Autonomous Region, Ningxia province, and Inner Mongolia Autonomous Region, were lower than the national average (Fig. 3). HCHO VCDs in central China were higher than the national average and the distribution of HCHO VCDs were homogenous. The Highest HCHO VCDs occurred in eastern China, especially in the Beijing-Tianjin-Hebei region, northern Henan province, western Shandong Province, the YRD region, and the Pearl River

Delta (PRD) region, and the most severe HCHO pollution occurred in the YRD. Living in the YRD with high outdoor HCHO concentration has a big risk of up to 155 in million to suffer cancer, which is about four times than that in Unite States (Zhu et al., 2017a). Total cancer risk in the YRD occupied 23.4 % of total national cancer risk and about 7840 people in the YRD may develop cancer in their lives due to severe HCHO pollution. It is urgent to control ambient HCHO concentration in the YRD effectively. Besides, the YRD has been identified as the highest VOCs emissions region (Qiu et al., 2014;Wu et al., 2015). So

determination of contributions from primary source and secondary formation to ambient HCHO is needed to formulate appropriate control measures.

In the YRD region, the highest HCHO VCDs occurred in southwestern Jiangsu province, e.g. Nanjing, Changzhou, Wuxi and Suzhou cities and northern Zhejiang province, e.g. Hangzhou, Jiaxing, Huzhou and Ningbo cities, and Shanghai in 2015 (Fig. 4a). The spatial distribution of annual mean HCHO VCDs in 2016 and 2017 were similar to that in 2015. HCHO VCDs

in the YRD region showed a fluctuating trend from 2015 to 2017 (Fig. 4b, c). HCHO concentration in Jiangsu province rose from 2015 to 2016, while only HCHO VCDs in southwestern Jiangsu province showed a decreasing trend from 2016 to 2017. HCHO concentration in Shanghai and Hangzhou showed a similar trend and decreased clearly from 2015 to 2016 and then increased from 2016 to 2017. An opposite trend of HCHO VCDs was observed in other cities of Zhejiang province.

**3.3 Determination of primary and secondary contributions to ambient HCHO**

The statistical analysis of simultaneous real-time measurements of HCHO, CO, and $O_3$ can be described by a multiple linear regression model:

$$C_{HCHO}=\beta_0+\beta_1\times C_{CO}+\beta_2\times C_{O_3} \tag{4}$$

where $\beta_0$, $\beta_1$, $\beta_2$ are the coefficients fitted by the model (Garcia et al., 2006), and $C_{CO}$ and $C_{O_3}$ represent the concentration of CO and $O_3$, respectively.

The relative contributions of background concentration, primary emission, and secondary formation to the ambient HCHO can be calculated through the following equations:





$$R_{Primary} = \frac{\beta_1 \times C_{CO}}{\beta_0 + \beta_1 \times C_{CO} + \beta_2 \times C_{O_3}} \times 100\ \% \tag{5}$$

$$R_{Secondary} = \frac{\beta_2 \times C_{O_3}}{\beta_0 + \beta_1 \times C_{CO} + \beta_2 \times C_{O_3}} \times 100\ \% \tag{6}$$

$$R_{Background} = \frac{\beta_0}{\beta_0 + \beta_1 \times C_{CO} + \beta_2 \times C_{O_3}} \times 100\ \% \tag{7}$$

where $R_{Primary}$ represents the contribution to ambient HCHO from primary sources, e.g. industrial and vehicle emissions; $R_{Secondary}$ represents the contribution to ambient HCHO from secondary sources, i.e. photochemical VOC oxidation); $R_{Background}$ represents background contributions to the ambient HCHO which can neither be classified to primary or secondary contribution. In previous studies in YRD (Wang et al., 2015;Ma et al., 2016), the background HCHO concentration of 1 ppbv was selected to represent the HCHO concentration in regional conditions. So the $\beta_0$ coefficient is fixed at 1 ppbv in the multiple linear regression.

In a previous study, Ma et al. (2016) analyzed the primary emission and secondary formation of HCHO from Apr 15[th] to May 1[st], 2015 at an industrial zoon in Nanjing which is located less than 15 km away from one of our research site, Maigaoqiao (abbr. MGQ) station, using in-situ measurement of primary tracers, i.e., benzene, toluene and CO and secondary tracer, i.e., $O_3$. Ma et al. (2016) found that the average relative contributions of industry-related emission and secondary formation and background were 59.2 %, 13.8 %, and 27 %, respectively. In this study, the average relative contributions from primary source, secondary formation and background to HCHO at MGQ station during the same period were 50.5 %, 30.8 % and 18.7 % respectively. We can see the relative contribution of primary emission was similar to that in Ma et al. (2016)'s study with the difference of 8.7 %. However, the relative contribution of secondary formation was 17 % larger than that in Ma et al. (2016)'s study and the relative contribution of background was 8.3 % smaller than that in Ma et al. (2016)'s study. It should be noted that the contribution of secondary formation we calculate is the maximum of a day because the overpass time of the OMPS is 13:30 (LT) every day. So it is reasonable that the contribution of secondary formation we calculate is larger due to more photochemical reaction at noon (Hong et al., 2018). In conclusion, tropospheric HCHO column observed by OMPS can be used to analyze primary and secondary contributions to ambient HCHO through multiple linear regression fit, and CO and $O_3$ concentrations measured by CNEMC Network can be used as the indicator of primary emission source and secondary formation source respectively.

### 3.3.1 Primary and secondary sources of ambient HCHO in Nanjing

The distribution of the three stations selected to explore the primary and secondary sources of HCHO are shown in Fig. 5a, and the heavy industry zone is circled by red lines (Zheng et al., 2015). In order to ensure the representativeness of regression model, only data fulfilling the criteria in the analysis which correlation coefficient (R) is larger than 0.6 and significance value is lower than 0.05 was kept, and the analysis for Hangzhou and Shanghai hereinafter is same. We finally select MGQ site located in the southwest of industrial zone, Ruijin road site (abbr. RJR) located in the center of Nanjing, and



site in Xianlin university town (abbr. XLUT) located in southeast of industrial zone to analyze HCHO concentration from different sources in Nanjing. As presented in Table 1, the annual average HCHO concentrations at industrial sites were larger than that at urban site in 2015 and 2017. The seasonal average HCHO concentration ranked in the order of summer > autumn > spring > winter, whether in industrial zone or center of Nanjing (Fig. 6 and Table 3).

On average, the secondary formation was the dominant source of ambient HCHO at all the three sites in Nanjing (Table 1). While primary emission may turn to be the most significant source in winter at RJR site (Fig. 6a, b), in winter of 2015 at MGQ site (Fig. 6c, d), and from Aug to Dec, 2015 at XLUT site (Fig. 6e, f). At industrial sites MGQ and XLUT, the annual average HCHO concentration from primary emission decreased by 64 % and 63.44 % from 2015 to 2017, respectively. Because coal combustion is an important primary source of ambient HCHO (Wang et al., 2013;Liu et al., 2017), energy savings and
energy consumption reducing, especially less consumption of coal, according to Nanjing Municipal government (http://www.nanjing.gov.cn/xxgk/szf/201403/t20140321_2544036.html, in Chinese), caused the diminishing of primary HCHO emission. The variation of HCHO concentration from different sources at MGQ was similar to that at XLUT, because of the short distance between the two sites. However, the annual average HCHO concentration from primary emission at XLUT was larger than that at MGQ. Primary emission of HCHO at urban site was smaller than that at industrial sites in 2015 and
larger than that at suburban sites in 2016 and 2017. The correlation coefficients between primary HCHO or secondary HCHO and total HCHO can be used to compare the contribution to the variation of HCHO from the two sources. At urban site, compared with primary emission (R=0.35, Fig. S2a), secondary formation contributed more to the variation of HCHO concentration from 2015 to 2017 (R=0.76, Fig. 7a). While at industrial sites, contribution from secondary formation to variation of HCHO concentration was less evident (R<=0.67, Fig. 7b, c). At XLUT site, primary HCHO emission was more significant
to variation of HCHO concentration than secondary formation in 2015 (Table 2). On the seasonal average, secondary formation contributed most to ambient HCHO concentration, and reached maximum in summer due to improvement of photochemical reaction (Wang et al., 2016) and reached minimum in winter (Table 3). In spring, summer and autumn, variation of ambient HCHO was affected more by secondary formation while in winter, it was influenced more by primary emission in urban region (Table 4).

### 3.3.2 Primary and secondary sources of ambient HCHO in Hangzhou

Three sites, including Xiasha (abbr. XS), Chengxiang Town (abbr. CXT), and The Fifth District of Chaohui (abbr. FDCH) which are located in the northeast suburbs, the eastern suburbs and the center of Hangzhou, respectively, were selected to explore the sources of HCHO in Hangzhou as shown in Fig. 5b. There was no significant difference between HCHO concentrations at suburban sites (Table 1). HCHO concentration at these three sites reached the maximum in summer and
reached the minimum in winter, which kept agreement with previous study (De Smedt et al., 2015). The average HCHO concentration in 2016 was larger than those in 2015 and 2017 at all these three sites.

As shown in Table 1, the secondary formation, on annual average, contributed more to ambient HCHO than primary





emission from 2015 to 2017 in Hangzhou. It is interesting that primary emission of HCHO showed a small increasing trend from 2015 to 2017 at suburban sites (Fig. 8a, c, and Table 1). It may be caused by the significant increase of number of vehicles and less effective control measures for ambient HCHO, according to the environment bulletin released by Hangzhou government. Secondary HCHO also kept an important growing tendency from 2015 to 2016 and turned to decrease in 2017.

And variation of HCHO concentration was mainly affected by secondary formation from 2015 to 2017, due to the strong correlation (R > 0.7) (Fig. 7d, e and f). In all seasons, seasonal average of secondary formation from 2015 to 2017 was larger than primary emission in Hangzhou (Table 3). While in winter of 2016 and in summer of 2017 at XS site (Fig. 8a, b), in winter of 2016 at CXT site (Fig. 8c, d), and in winter of 2016 and 2017 at urban site (Fig. 8e, f), primary emission may exceed secondary formation and become the main source in individual days. In winter, the impact of primary emission on variation of

HCHO was more significant, compared with secondary formation. While in spring, summer and autumn, secondary formation became more important to variation of ambient HCHO, except in the northeast suburbs in autumn (Table 4).

### 3.3.3 Primary and secondary sources of ambient HCHO in Shanghai

Site of Pudong New Area (abbr. PDNA), Hongkou site (abbr. HK), Dianshan Lake in Qingpu (abbr. DSL) were selected to represent east of the city center, north of the city center, and suburb, respectively (Fig. 5c). The annual average HCHO

concentration decreased by 0.342 ppbv from 2015 to 2016 and increased by 0.529 ppbv from 2016 to 2017 at HK site (Table 1). And HCHO concentration at PDNA site increased by 0.15 ppbv in 2016 and decreased by 0.056 ppbv in 2017, compared to that in previous year. At DSL sites, HCHO concentration was less than that at HK, except in 2016. And HCHO concentration firstly increased by 0.179 ppbv from 2015 to 2016, and then decreased by 0.143 ppbv from 2016 to 2017. We can conclude that ambient HCHO concentration was not controlled strictly in recent years. On the seasonal average, HCHO concentration

reached its maximum in summer and minimum in winter in Shanghai which is same to that in Nanjing and Hangzhou (Table 3). Besides, HCHO concentration in summer in Shanghai was larger than that in Nanjing and Hangzhou.

On average, secondary formation was the most significant contribution to ambient HCHO in Shanghai, except at PDNA site in 2016 (Table 1 and Fig. 9). The contribution from secondary formation developed with a rising trend and primary HCHO emission showed a decreasing trend from 2015 to 2017, at suburban site (Fig. 9e, f and Table 1). Variation of HCHO

concentration was affected more largely by secondary formation (R>0.70, Fig. 7g, h and i) than primary emission (R<0.43, Fig. S2g, h, and i) from 2015 to 2017. In summer and autumn, secondary formation was the largest source to ambient HCHO (Fig. 9a, b, c, and d) and was the main factor of variation of HCHO concentration in urban regions of Shanghai. While at PDNA site, primary emission became the most important source to ambient HCHO in spring and winter of 2016 and influenced more significantly than secondary formation on variation of ambient HCHO in winter.



## 4 Discussions

### 4.1 Reconsidering of the proxy for VOCs reactivity using HCHO concentration

Duncan et al. (2010) determined the indicator of VOCs and NOx controls on surface $O_3$ formation using HCHO VCDs observed by OMI as the proxy for VOCs reactivity in august between 2015 and 2007 in the Unites States. It is reasonable because ambient HCHO concentration is mainly from the atmospheric oxidation of isoprene in summer over North America (Palmer et al., 2003;Zhu et al., 2017b;Marvin et al., 2017) and HCHO concentration from secondary formation contributed 87 percent during daytime in Los Angeles in 1980s (Kawamura et al., 2000) and over 80 percent ambient HCHO concentration is from secondary formation over the Eastern American (Luecken et al., 2006). However, contribution from secondary formation in the YRD is smaller than that in America, accounting for about 70 percent in summer and 40-54 percent in winter in the YRD. Total HCHO can be regarded as the proxy for VOCs reactivity over a three-year study period from 2015 to 2017 because of the good correlation between total HCHO with secondary HCHO in Nanjing, Hangzhou, and Shanghai (R >= 0.59). If the study period is only in 2015, using ambient HCHO concentration to indicate VOCs reactivity would cause significant errors because ambient HCHO showed a stronger correlation with primary emission than secondary formation, for example, in the industrial zone of Nanjing and in the suburb of Shanghai. Besides, the correlation between total HCHO and secondary HCHO depends on seasons. In winter, ambient HCHO concentration cannot be represented by that from secondary formation due to the poor correlation (R<0.12, P>0.05). In spring, at RJR and DSL sites, and in summer, at RJR, CXT, HK, and PDNA sites, and in autumn, at RJR and MGQ sites, total HCHO can be used as the proxy for VOCs reactivity (R>0.59, P<0.01).

### 4.2 HCHO control measures adapted to local conditions

Sources of ambient HCHO at different sites varied largely and it is closely related to different industrial structure (Zheng et al., 2016). In order to make HCHO control measures adapted to local conditions better, it is meaningful to estimate different sources of ambient HCHO. Results in Sect. 3.3 indicate that the main influencing factor of ambient HCHO concentration depends to seasons. In spring, summer, and autumn, controlling VOCs emissions strictly has a positive and sensitive effect on reducing ambient HCHO concentration in Shanghai and urban areas of Nanjing and Hangzhou. However, in winter, decreasing primary emission of HCHO contributes to reducing ambient HCHO concentration in Nanjing, Shanghai and urban areas of Hangzhou. In urban regions with heavy traffic, controlling the number of private cars and improving traffic management not only contribute to decreasing primary emission of HCHO but also help decrease VOCs concentration (Wang and Zhao, 2008;Liu et al., 2017).

The industries of Nanjing include chemical industry, steel plant, and cement industry (Zheng et al., 2016). So at industrial sites, decreasing HCHO emissions from related-chemical activities is an efficient way to control ambient HCHO concentration in winter. In Hangzhou, labor-intensive industries are most important parts of industry economics (Zheng et al., 2016). The largest amount of VOCs emissions were measured in 2015 in the eastern suburbs of Hangzhou and VOCs emissions from



furniture manufacturing is the largest sources (Lu et al., 2018a). However, industrial manufacturing is developing well and VOCs emissions occur in many types of industries i.e., textile industry, printing, Leather manufacturing and shoemaking (Liu et al., 2008;Khan and Malik, 2014). It is a huge challenge to control VOCs emissions in this region. Unlike Nanjing and Hangzhou, tertiary industry has played a leading roles in Shanghai's economic since 2013 and primary and secondary industries have decreased since 1999 (Chen et al., 2016). However, HCHO concentration from primary and secondary sources was not controlled effectively. Decreasing private cars on road and developing evaporative control regulations suited for local conditions help reduce HCHO not only from primary emission but also from oxidation from VOCs, especially in downtown (Liu et al., 2015).

## 5 Summary

Tropospheric HCHO VCDs derived from OMPS observations were validated using ground-based FTS measurements in Hefei, from 2015 to 2017. HCHO VCDs observed by OMPS and FTS are in good agreement (R=0.78). Serious HCHO pollution was observed in the YRD region, especially in southwestern Jiangsu province, e.g. Nanjing, Changzhou, Wuxi and Suzhou cities and northern Zhejiang province, e.g. Hangzhou, Jiaxing, Huzhou and Ningbo cities, and Shanghai. Estimating different sources of ambient HCHO and then formulating control measures adapted to local conditions are effective ways to control HCHO pollution. However, measurements of HCHO are scarce, both spatially and temporally. In this study, we analyzed primary and secondary contributions to ambient HCHO using multiple linear regression based on HCHO observed by OMPS. At MGQ site in Nanjing, the average relative contributions from primary emission, secondary formation and background to HCHO from Apr. 15th to May. 1th, 2015 were 50.5 %, 30.8 % and 18.7 %, respectively, which kept good agreement with previous study. So tropospheric HCHO columns observed by OMPS can be used to estimate different sources of ambient HCHO.

Primary and secondary contributions to ambient HCHO in Nanjing, Hangzhou, and Shanghai from 2015 to 2017 were determined. Overall, HCHO concentration from secondary formation contributed most to ambient HCHO in the megacities in the YRD region and influenced variation of ambient HCHO concentration in Hangzhou, Shanghai, and urban region of Nanjing. While at industrial sites of Nanjing, the annual average HCHO concentration from primary emission decreased largely from 2015 to 2016 due to energy savings and energy consumption reducing and contribution from secondary formation to variation of HCHO concentration was less significant. At suburban regions of Hangzhou, primary emission showed a small increasing trend from 2015 to 2017. And at urban sites, primary emission of HCHO in Shanghai was larger than that in Hangzhou. Seasonally, secondary HCHO reached the maximum in summer and minimum in winter. In spring, summer, and autumn, secondary formation played a crucial role in variation of HCHO concentration in urban areas of Nanjing, Hangzhou and Shanghai. While in winter, primary emission contributed more to variation of ambient HCHO in Nanjing, Shanghai, and urban areas of Hangzhou. These findings contribute to formulating effective HCHO pollution control measures. Besides, only secondary HCHO can be used as the proxy for VOCs reactivity. So separating HCHO from different sources is full of great



significance in sensitivity studies of O$_3$ production.

*Author contributions.* The first two authors contributed equally. CL and QH designed and supervised the study and CL also help retrieve the satellite data and WS wrote the manuscript. SZ, YS, and WW retrieved the FTS data, and YZ ran the WRF-Chem model. JL supported the project and JK contributed to discuss the results.

*Acknowledgments.* This work was supported by National Key Research and Development Program of China (no. 2017YFC0210002, 2018YFC0213104, 2018YFC0213100, 2016YFC0203302), and National Natural Science Foundation of China (no. 41722501, 91544212, 51778596, 41575021), and the National High-Resolution Earth Observation Project of China (grant no 05-Y20A16-9001-15/17-2). The authors acknowledge OMPS project for making OMPS-NPP Nadir Mapper Earth View Level 1B data product available on the Internet. We also thank Ministry of Environment Protection of the People's

Republic of China for making data measured by atmospheric environment automatic monitoring stations available on the internet.

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



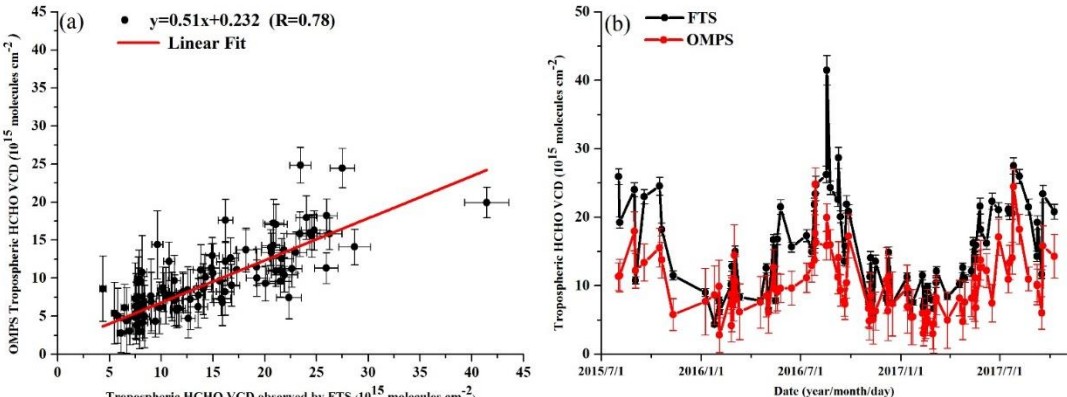

**Figure 1. (a) Correlation analysis and (b) time series of tropospheric HCHO VCDs measured by OMPS and FTS from Jul.1st, 2015 to Nov.20th, 2017.**

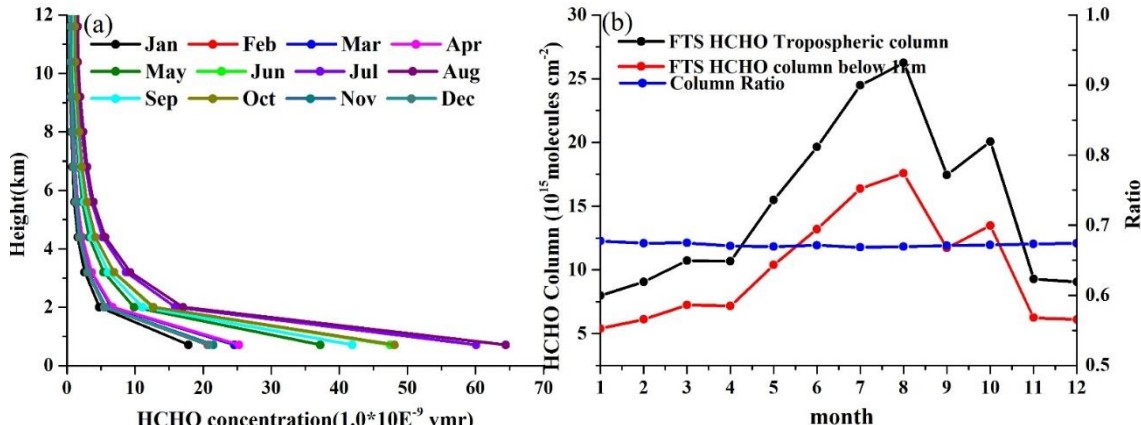

**Figure 2. (a) The monthly averaged HCHO vertical profiles in the troposphere measured by FTS, and (b) the monthly averaged tropospheric HCHO column (dark line) and HCHO column below 1 km (red line) measured by FTS and the ratio of HCHO column below 1km in the tropospheric HCHO column (blue line).**



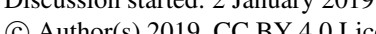


**Figure 3. The spatial distribution of monthly averaged tropospheric HCHO VCDs observed by OMPS and cancer risks in August, 2017.**





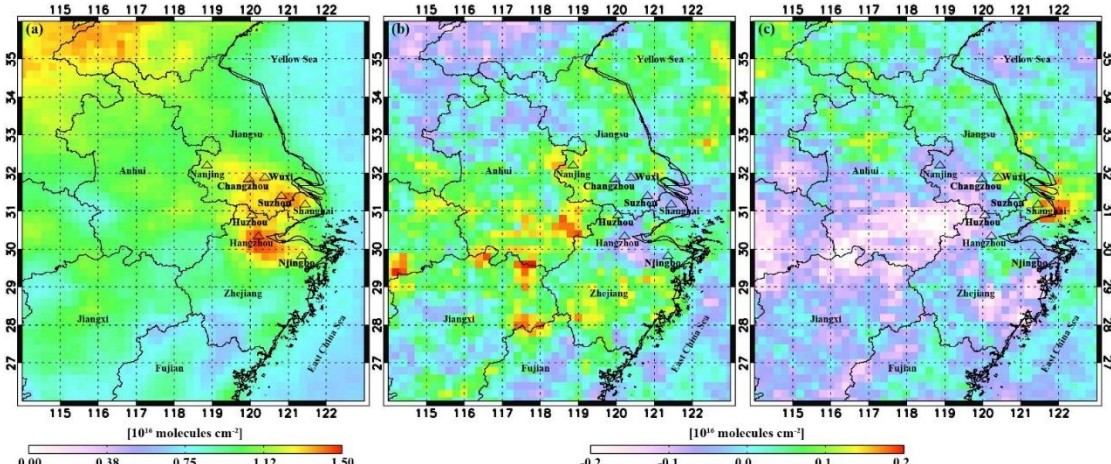

**Figure 4. (a) The spatial distribution of annual mean tropospheric HCHO Colum observed by OMPS through YRD in 2015 and (b) changes in the HCHO VCDs between 2016 and 2015 and (c) between 2017 and 2016.**



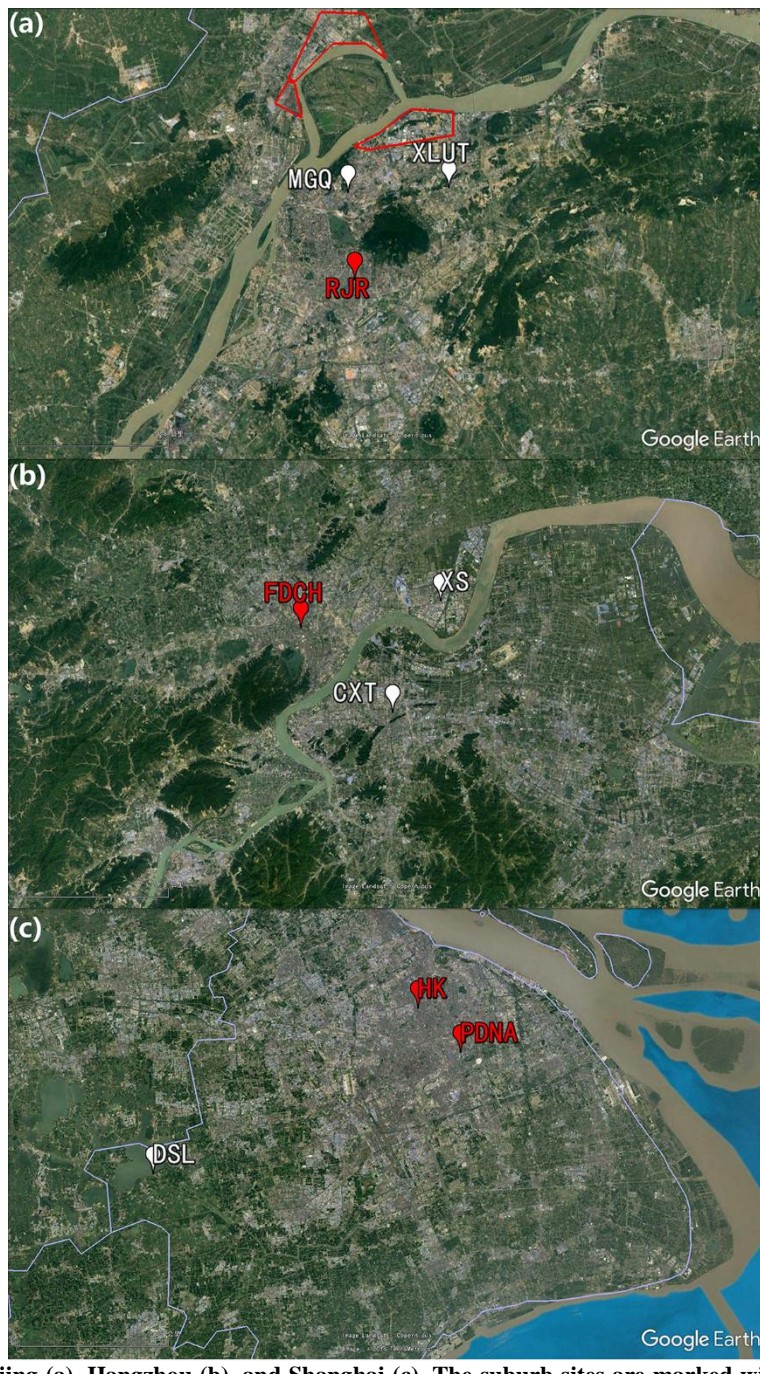

**Figure 5. The maps of Nanjing (a), Hangzhou (b), and Shanghai (c). The suburb sites are marked with white, and the downtown sites are marked with red. And the industrial zones are circled by red lines.**





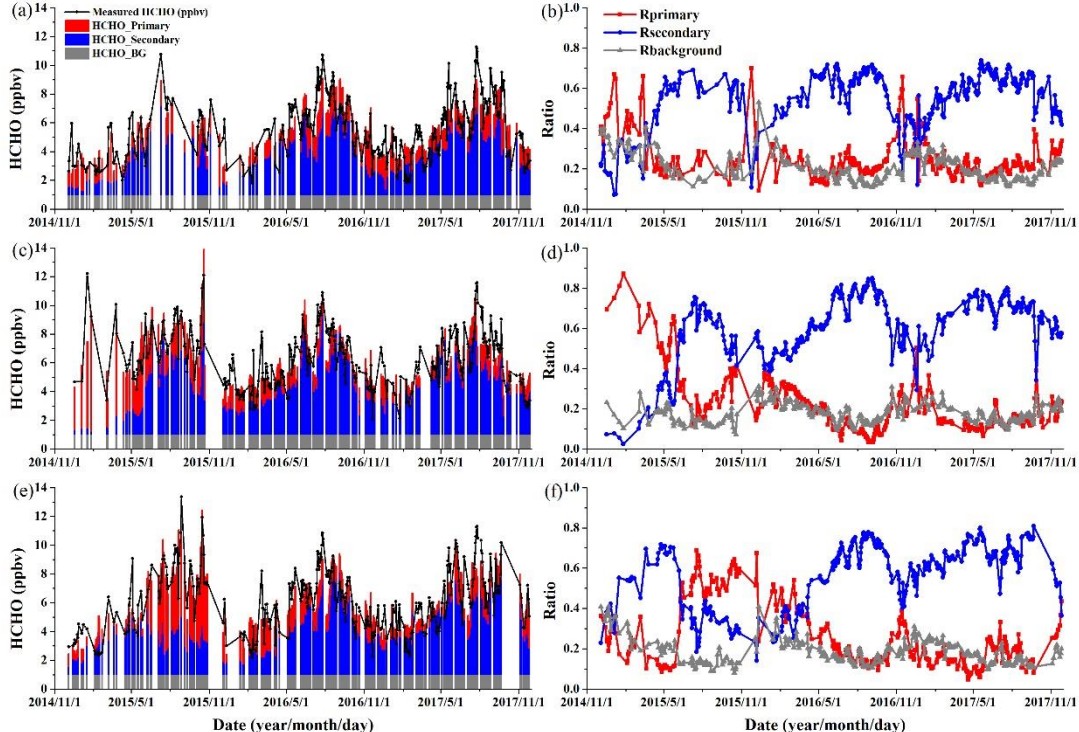

**Figure 6. The time series of absolute and relative contributions of primary and secondary sources and background to HCHO concentration from Dec 2014 to Nov 2017 at RJR site (a, b), MGQ site (c, d), and XLUT site (e, f) in Nanjing.**



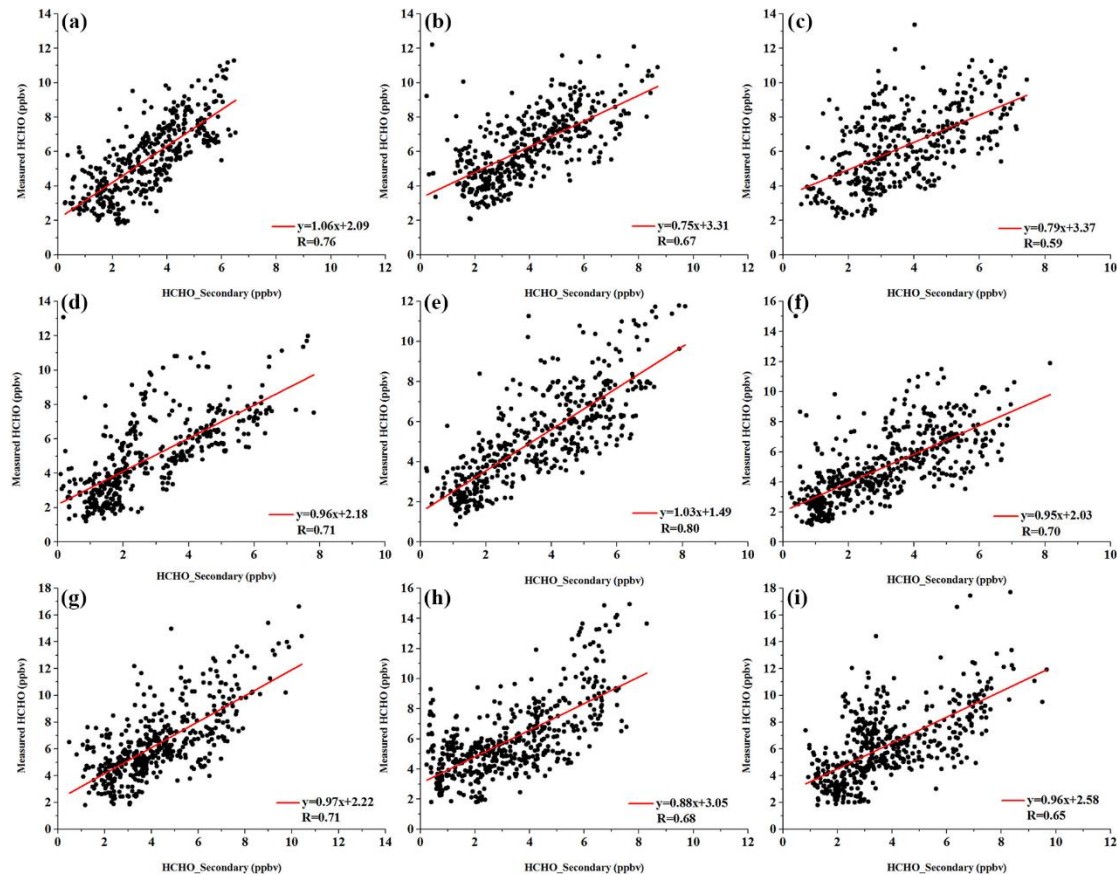

Figure 7. Correlation analysis of total HCHO and secondary HCHO from 2015 to 2017 at RJR site (a), MGQ site (b), and XLUT site (c) in Nanjing, at XS site (d), CXT site (e), and FDCH site (f) in Hangzhou, and at HK site (g), PDNA site (h), and DSL site (i) in Shanghai.



**Table 1. Annual average of measured HCHO concentrations (ppbv) and absolute and relative contributions from different sources in Nanjing, Hangzhou, and Shanghai.**

|  |  | Nanjing | | | Hangzhou | | | Shanghai | | |
|---|---|---|---|---|---|---|---|---|---|---|
| year | | MGQ (industry zone) | RJR (urban area) | XLUT (industry zone) | XS (suburb) | CXT (suburb) | FDCH (urban area) | HK (urban area) | PDNA (urban area) | DSL (suburb) |
| 2015 | HCHO primary | 2.528 (36.86 %) | 1.288 (18.16 %) | 2.853 (14.63 %) | 0.523 (11.26 %) | 0.333 (8.76 %) | 0.295 (7.23 %) | 1.956 (31.08 %) | 1.155 (20.24 %) | 2.602 (37.15 %) |
| | HCHO Secondary | 3.556 (48.13 %) | 2.342 (46.41 %) | 2.601 (42.80 %) | 2.892 (62.20 %) | 3.258 (65.40 %) | 3.483 (68.13 %) | 3.497 (52.66 %) | 3.681 (61.68 %) | 2.535 (43.71 %) |
| | HCHO | 7.205 | 4.68 | 6.506 | 4.434 | 4.656 | 4.817 | 6.422 | 5.822 | 6.145 |
| 2016 | HCHO Primary | 0.933 (17.75 %) | 1.411 (11.49 %) | 1.328 (23.87 %) | 1.154 (27.57 %) | 0.768 (14.55 %) | 1.981 (37.22 %) | 0.792 (13.08 %) | 2.348 (42.52 %) | 1.015 (17.10 %) |
| | HCHO Secondary | 3.948 (63.88 %) | 3.804 (59.26 %) | 3.776 (58.58 %) | 3.053 (50.84 %) | 4.033 (66.03 %) | 2.418 (41.56 %) | 4.275 (68.51 %) | 3.688 (39.15 %) | 4.309 (65.40 %) |
| | HCHO | 5.945 | 6.249 | 6.14 | 5.137 | 5.356 | 5.294 | 6.08 | 5.972 | 6.324 |
| 2017 | HCHO Primary | 0.910 (15.39 %) | 1.209 (10.19 %) | 1.043 (16.48 %) | 2.233 (25.84 %) | 0.687 (14.53 %) | 0.511 (12.03 %) | 0.816 (13.19 %) | 1.408 (25.72 %) | 0.945 (15.21 %) |
| | HCHO Secondary | 4.264 (67.50 %) | 3.303 (57.76 %) | 4.209 (66.18 %) | 1.823 (50.66 %) | 3.276 (62.36 %) | 3.112 (62.74 %) | 4.823 (70.14 %) | 3.591 (56.16 %) | 4.259 (66.55 %) |
| | HCHO | 6.205 | 5.461 | 6.289 | 5.022 | 4.926 | 4.626 | 6.609 | 5.916 | 6.181 |

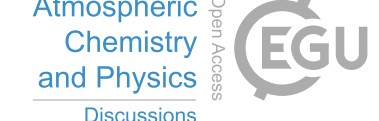



**Table 2. The Pearson correlation coefficient (R) between the measured HCHO and primary HCHO emission and secondary HCHO in Nanjing, Hangzhou, and Shanghai from 2015 to 2017.**

| year | | Nanjing | | | Hangzhou | | | Shanghai | |
|---|---|---|---|---|---|---|---|---|---|
| | | MGQ (industry zone) | RJR (urban area) | XLUT (industry zone) | XS (suburb) | CXT (suburb) | FDCH (urban area) | HK (urban area) | PDNA (urban area) | DSL (suburb) |
| **2015** | HCHO primary | 0.14 | 0.21 | 0.84 | 0.69 | -0.1 | 0.06 | 0.18 | 0.2 | 0.83 |
| | HCHO Secondary | 0.58 | 0.75 | 0.47 | 0.9 | 0.82 | 0.81 | 0.55 | 0.65 | 0.61 |
| **2016** | HCHO Primary | -0.43 | 0.27 | -0.3 | -0.4 | 0.31 | 0.51 | 0.42 | 0.06 | 0.28 |
| | HCHO Secondary | 0.82 | 0.71 | 0.73 | 0.77 | 0.84 | 0.68 | 0.77 | 0.69 | 0.81 |
| **2017** | HCHO Primary | 0.07 | 0.5 | 0.51 | 0.84 | 0.42 | 0.38 | 0.26 | 0.06 | 0.58 |
| | HCHO Secondary | 0.7 | 0.77 | 0.74 | 0.68 | 0.76 | 0.79 | 0.77 | 0.75 | 0.74 |





**Table 3. Seasonal average of measured HCHO concentrations (ppbv) and absolute and relative contributions from different sources from 2015 to 2017 in Nanjing, Hangzhou, and Shanghai.**

| season | | Nanjing | | | Hangzhou | | | Shanghai | | |
| --- | --- | --- | --- | --- | --- | --- | --- | --- | --- | --- |
| | | MGQ (industry zone) | RJR (urban area) | XLUT (industry zone) | XS (suburb) | CXT (suburb) | FDCH (urban area) | HK (urban area) | PDNA (urban area) | DSL (suburb) |
| winter | HCHO primary | 1.398 (29.45%) | 1.151 (32.01%) | 1.013 (25.15%) | 0.866 (25.58%) | 0.46 (16.50%) | 0.876 (25.77%) | 1.032 (22.25%) | 2.383 (50.13%) | 0.741 (20.72%) |
| | HCHO Secondary | 2.011 (47.15%) | 1.469 (39.54%) | 1.993 (48.73%) | 1.209 (40.75%) | 1.332 (46.82%) | 1.061 (38.48%) | 2.197 (53.13%) | 1.212 (27.54%) | 1.874 (51.17%) |
| | HCHO | 4.403 | 3.394 | 3.694 | 2.627 | 2.342 | 2.491 | 3.841 | 4.098 | 3.338 |
| spring | HCHO Primary | 1.51 (27.04%) | 0.981 (20.00%) | 0.922 (18.14%) | 0.606 (15.89%) | 0.44 (10.71%) | 0.645 (16.59) | 0.728 (14.30%) | 1.598 (35.00%) | 0.589 (12.70%) |
| | HCHO Secondary | 3.017 (54.37%) | 2.996 (59.04%) | 3.365 (62.45%) | 2.352 (57.94%) | 2.744 (64.27%) | 2.288 (57.21%) | 3.399 (65.71%) | 2.043 (42.88%) | 3.166 (65.83%) |
| | HCHO | 5.619 | 5.161 | 5.575 | 4.259 | 4.473 | 4.193 | 5.334 | 4.946 | 4.975 |
| summer | HCHO Primary | 1.047 (13.89%) | 1.414 (20.64%) | 1.964 (25.53%) | 1.878 (25.59%) | 0.695 (10.26%) | 1.139 (16.86%) | 1.051 (13.75%) | 1.526 (19.98%) | 2.013 (24.31%) |
| | HCHO Secondary | 5.503 (72.46%) | 4.484 (64.54%) | 4.542 (60.70%) | 4.389 (60.45%) | 5.143 (74.66%) | 4.512 (67.79%) | 5.875 (72.97%) | 5.273 (66.66%) | 5.361 (63.41%) |
| | HCHO | 7.764 | 7.277 | 7.919 | 7.451 | 7.189 | 6.936 | 8.36 | 8.196 | 8.602 |
| autumn | HCHO Primary | 1.103 (18.36%) | 1.537 (26.40%) | 1.99 (28.31%) | 1.485 (20.11%) | 0.775 (14.86%) | 0.982 (17.48%) | 1.538 (23.46%) | 1.529 (22.93%) | 2.231 (30.91%) |
| | HCHO Secondary | 3.905 (63.95%) | 3.462 (55.73%) | 3.921 (56.39%) | 4.478 (64.87%) | 4.163 (66.62%) | 3.876 (64.08%) | 4.241 (61.15%) | 4.163 (61.65%) | 3.87 (54.57%) |
| | HCHO | 5.851 | 5.463 | 6.352 | 6.654 | 5.496 | 5.477 | 6.183 | 5.937 | 6.789 |

  

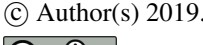



**Table 4. The Pearson correlation coefficient (R) between the measured HCHO and primary HCHO emission and secondary HCHO in Nanjing, Hangzhou, and Shanghai in winter, spring, summer, and autumn.**

| season | | Nanjing | | | Hangzhou | | | Shanghai | |
|---|---|---|---|---|---|---|---|---|---|
| | | MGQ (industry zone) | RJR (urban area) | XLUT (industry zone) | XS (suburb) | CXT (suburb) | FDCH (urban area) | HK (urban area) | PDNA (urban area) | DSL (suburb) |
| winter | HCHO primary | 0.67 | 0.24 | -0.05 | 0.42 | 0.01 | 0.47 | 0.61 | 0.61 | 0.5 |
| | HCHO Secondary | -0.41 | -0.23 | 0.12 | -0.24 | 0.03 | -0.25 | -0.42 | -0.42 | -0.44 |
| spring | HCHO Primary | 0.11 | 0.21 | 0.17 | 0.04 | 0.29 | 0.13 | 0.12 | 0.16 | -0.21 |
| | HCHO Secondary | 0.25 | 0.6 | 0.43 | 0.43 | 0.47 | 0.26 | 0.56 | 0.39 | 0.62 |
| summer | HCHO Primary | 0.17 | 0.54 | 0.22 | 0.08 | -0.02 | 0.2 | 0.1 | 0.34 | 0.08 |
| | HCHO Secondary | 0.45 | 0.59 | 0.26 | 0.33 | 0.69 | 0.35 | 0.6 | 0.62 | 0.35 |
| autumn | HCHO Primary | 0.42 | 0.26 | 0.54 | 0.71 | 0.05 | -0.03 | 0.26 | 0.36 | 0.31 |
| | HCHO Secondary | 0.71 | 0.68 | 0.37 | 0.12 | 0.48 | 0.55 | 0.41 | 0.48 | 0.19 |





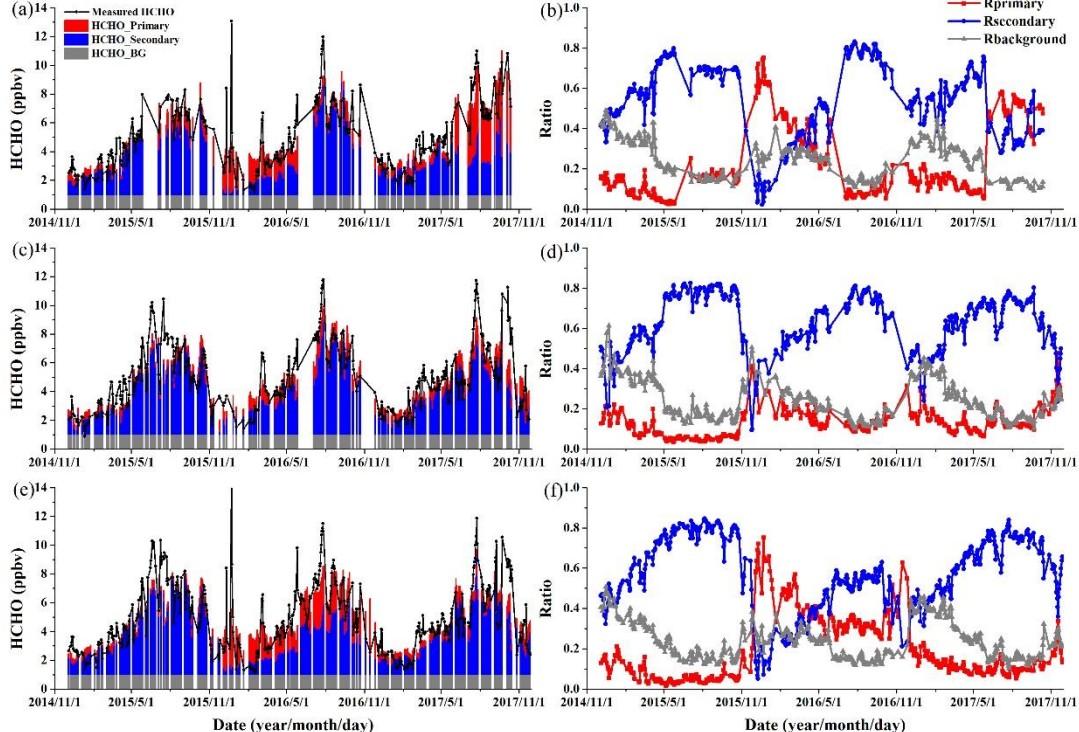

**Figure 8. The time series of absolute and relative contributions of primary and secondary sources and background to HCHO concentration from Dec 2014 to Nov 2017 at XS (a, b), CXT (c, d), and FDCH (e, f) sites in Hangzhou.**





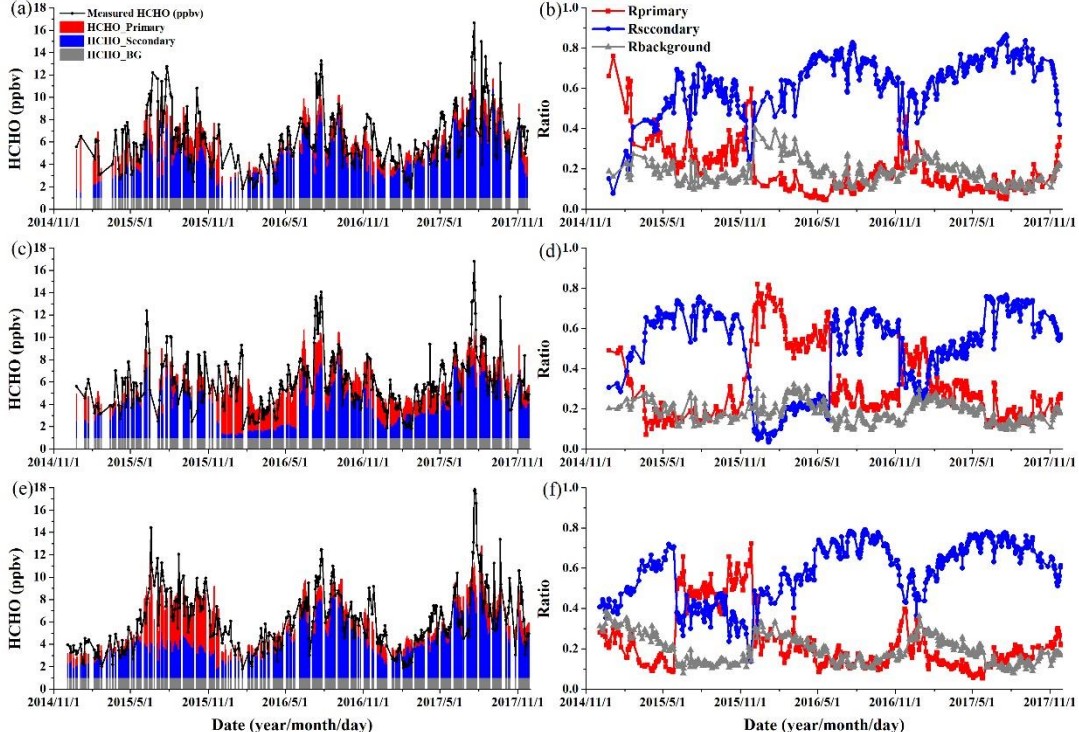

**Figure 9. The time series of absolute and relative contributions of primary and secondary sources and background to HCHO concentration from Dec 2014 to Nov 2017 at HK (a, b), PDNA (c, d), and DSL (e, f) sites in Shanghai.**

