# Peer review of "Primary and secondary sources of ambient formaldehyde in the Yangtze River Delta based on OMPS observation"

_Atmospheric Chemistry and Physics, 2018_

## Referee Comment (RC1) · Anonymous Referee #2 · 7 Jan 2019

Source apportionment of primary and secondary HCHO is a meaning work to understand the atmospheric chemistry of HCHO and control HCHO pollution. The authors explore the source of HCHO in the YRD using the satellite HCHO and $O_3$ and CO from CNEMC network following multiple linear regression analysis. I recommend publication with reference to the following comments.

1. The authors use HCHO from satellite and primary tracer CO and secondary tracer O3 from ground in-situ measurements to explore the primary and secondary source of HCHO. HCHO data is column density while O3 and CO are concentrations at the surface. How the authors link the two different data to determine the source of HCHO. Why not use O3 and CO data from satellite?

2. Section 3.1: The authors compared the OMPS data with FTS data, but did not illustrate why to compare the two different dataset.

3. The authors discussed whether primary emission or secondary formation contributed significantly to the variations of HCHO, e.g. L15-20, P9, L5-10, P10, but did not tell the reader how they determine the contribution of primary and secondary sources to the variations of HCHO. The authors need a standard to quantify the contribution to the variations of HCHO.

4. The author discussed the primary and secondary sources of HCHO in the three megacity Shanghai, Nanjing and Hangzhou, e.g. the spatial and seasonal variations. They should also discuss the commonness and difference of HCHO in the three cities.

5. Sec. 4.2: The discussion of HCHO control measures should use more results from this study instead of just citing the references.

6. In sec. 5 summary: The authors declare "only secondary HCHO can be used as the proxy for VOCs reactivity", however, in sec. 4.1, the authors pointed out "Total HCHO can be regarded as the proxy for VOCs reactivity over a three-year study period". It seems paradoxical. The authors should make it explicit.

7. So separating HCHO from different sources is full of great significance insensitivity studiesof O3production.

8. Line 12 in Page 2: "causing" should be "cause".

9. Line 13 in Page 2: "favor to" should be "be in favor of".

10. Line 21 in Page 5: "Surface air pollutants monitored by CNEMC was provided" should be "Surface air pollutants monitored by CNEMC were provided"

11. Line 10 in Page 7: "the distribution of HCHO VCDs were homogenous "should be "the distribution of HCHO VCDs was homogenous ".

12. Line 28 in Page 8: "The distribution of the three station selected to explore the primary and secondary sources of HCHO are shown" should be "The distribution of the three station selected to explore the primary and secondary sources of HCHO is shown"

13. Line 4 in Page 11: The year "2015" should be "2005". The paper published in 2010 cannot research O3 pollution in 2015.

14. Figure 3, the label of Yellow Sea and Bohai Sea goes against each other.

15. Figure 7: what are the P-value from the significance tests?

---

## Referee Comment (RC2) · Anonymous Referee #1 · 26 Jan 2019

The manuscript "Primary and secondary sources of ambient formaldehyde in the Yangtze River Delta based on OMPS observation" by Wenjing Su et al. apportions the primary and secondary sources of ambient HCHO using satellite observation, and discussed the application of HCHO to the study of tropospheric ozone production sensitivity. The paper is a significant exploration to obtain the spatiotemporal and source information of HCHO, and the results are believable. Some expression and discussion should be improved prior to publication.

Specific comments: Page 6, section 3.2: the author estimated the total national cancer risk of 33500 people and 439 cancer cases per year using OMPS HCHO observation.

[Figure]

Please explain how to convert?

Page 6, Line 10-11: HCHO concentrations from OMPS measurement were generally lower than those from FTS, and the underestimation from OMPS was attributed to errors of spectral fitting and AMF calculation. The authors should explain the detailed reasons of errors instead of simply attributing spectral fitting and AMF calculation.

Page 10, section 3.3.3 line 28-30: The authors conclude that primary emission influenced the variation of ambient HCHO more significantly than secondary formation in winter. This conclusion should be supported by quantitative analysis.

Page 11, section 4.1: The author discussed whether total HCHO can be regarded as the proxy for VOCs reactivity depending on the correlation between total HCHO with secondary HCHO. Why the relationship between them can be used to judge the representation of HCHO as the proxy for VOC reactivity?

Page 11, section 4.2: When discussing the HCHO control measures, it should be focused on HCHO pollution, i.e., HCHO concentrations beyond the air quality standard. When HCHO concentration is low, it is not necessary to discuss whether paying more attention to primary emission or secondary formation

Technical corrections: Page2, Line4: change "become increasing serious" to "become increasingly serious" Page5, Line20: change "Surface air pollutants monitored by CNEMC was" to "Surface air pollutants monitored by CNEMC were" Page5, Line27: change "one of the condition" to "one of the conditions" Page8, Line11: change "industrial zoon" to "industrial zone"

---

## Author Comment (AC1) · 13 Mar 2019

Response to acp-2018-1192-RC1

We really appreciate your constructive comments and suggestions on our manuscript. Your positive advice help improve our manuscript. We have considered every comment carefully, and responded on a point to point and marked every change in red in the revised version.

1. The authors use HCHO from satellite and primary tracer CO and secondary tracer O3 from ground in-situ measurements to explore the primary and secondary source of HCHO. HCHO data is column density while $O_3$ and CO are concentrations at the surface. How the authors link the two different data to determine the source of HCHO. Why not use O3 and CO data from satellite?

**Responses:** Thank you very much for this suggestion. HCHO column was converted to the mixing ratio at the surface using the formula (3) in the manuscript, and detailed information was described in sec 3.1. CO can also be observed by Atmospheric InfraRed Sounder (AIRS) on board satellite EOS-Aqua and Tropospheric Emission Spectrometer (TES) on board satellite EOS-Aura, close in time to OMPS observation. However, the global coverage of TES is significantly lower and the AIRS instrument is degrading (Worden et al., 2013). $O_3$ and CO concentrations at the surface were measured simultaneously from China National Environmental Monitoring Center (CNEMC) Network in long time series from 2014. So we use O3 and CO data at the surface from CNEMC Network instead of data from satellite to explore the sources of HCHO.

**Changes in manuscript:** L33-34, P3 to L1-3, P4 in the revised version: "Due to low global coverage of Tropospheric Emission Spectrometer (Aura-TES) (Luo et al., 2002; Luo et al., 2007) and increasing noise after 2004 of Atmospheric InfraRed Sounder (Aqua-AIRS) (Pagano et al., 2012) which measure tropospheric CO closely in time to OMPS observation, obtaining precise CO concentration in the YRD after 2013 from satellite was limited. Therefore, CO and O3 concentrations at the surface monitored by the China National Environmental Monitoring Center (CNEMC) Network simultaneously were used in this study."

2. Section 3.1: The authors compared the OMPS data with FTS data, but did not illustrate why to compare the two different dataset.

**Responses:** Thank you very much for this suggestion. The global Network for Detection of Atmospheric Composition Change (NDACC) provides high quality, consistent, and standardized measurements of trace gases using FTS instruments, and its retrieval results were widely used for satellite observation. The ground-based FTS instrument used in this study is a candidate site for NDACC, and long-term measurements of HCHO by FTS were performed. So HCHO concentration measured by FTS was used to validate OMPS HCHO observation.

**Changes in manuscript:** L6, P6 in the revised version: "In order to validate the OMPS observation with FTS measurement".

3. The authors discussed whether primary emission or secondary formation contributed significantly to the variations of HCHO, e.g. L15-20, P9, L5-10, P10, but did not tell the reader how they determine the contribution of primary and secondary sources to the variations

of HCHO. The authors need a standard to quantify the contribution to the variations of HCHO.

**Responses:** Thank you very much for this suggestion. Primary emission and secondary formation were fitted linearly with total HCHO, separately. And larger correlation coefficient means more contribution to the variations of HCHO.

**Changes in manuscript:** L27-28, P9, in the revised version: "and larger correlation coefficient means more contribution to the variations of HCHO".

4. The author discussed the primary and secondary sources of HCHO in the three megacity Shanghai, Nanjing and Hangzhou, e.g. the spatial and seasonal variations. They should also discuss the commonness and difference of HCHO in the three cities.

**Responses:** Thank you very much for this suggestion. The secondary source of HCHO in the three megacity have similar seasonal variation, ranking in the order of summer > autumn > spring > winter, similar to seasonal variation of HCHO concentration. And HCHO concentration form secondary formation in summer in Shanghai was larger than that in Nanjing and Hangzhou.

Changes in manuscript: L10-12, P11, in the revised version: "The secondary source of HCHO in the three megacity has similar seasonal variation, ranking in the order of summer > autumn > spring > winter, similar to seasonal variation of HCHO concentration. While HCHO concentration form secondary formation in summer in Shanghai was larger than that in Nanjing and Hangzhou."

5. Sec. 4.2: The discussion of HCHO control measures should use more results from this study instead of just citing the references.

**Responses:** Thank you very much for this suggestion.

**Changes in manuscript:** L18-20, P12, in the revised version: "In the Nanjing industrial zone, secondary source of HCHO was about four times larger than primary source in 2017. So at industrial sites, decreasing VOC emissions from related-chemical activities is an efficient way to control ambient HCHO concentration."; L21-27, P12, in the revised version: "Secondary HCHO in the eastern suburbs of Hangzhou was the largest from 2016 to 2017 (Table 1), corresponding to the previous study indicating that the largest amount of VOCs emissions were measured in the eastern suburbs of Hangzhou and VOCs emissions from furniture manufacturing is the largest sources (Lu et al., 2018a). Secondary HCHO contributed most to ambient HCHO, while was not controlled effectively from 2015 to 2017. Considering that industrial manufacturing is developing well and VOCs emissions occur in many types of industries i.e., textile industry, printing, Leather manufacturing and shoemaking (Liu et al., 2008; Khan and Malik, 2014), controlling VOCs emissions in Hangzhou will be a huge challenge."; L29-32, P12, in the revised version: "However, HCHO concentration from secondary source was not controlled effectively. Decreasing private cars on road and developing evaporative control regulations and improving traffic management suited for local conditions help reduce not only VOCs emissions but also HCHO emissions, especially in downtown (Wang and Zhao, 2008; Liu et al., 2015; Liu et al., 2017)."

6. In sec. 5 summary: The authors declare "only secondary HCHO can be used as the proxy

for VOCs reactivity", however, in sec. 4.1, the authors pointed out "Total HCHO can be regarded as the proxy for VOCs reactivity over a three-year study period". It seems paradoxical. The authors should make it explicit.

**Responses:** Thank you very much for this suggestion. It is corrected in the revised version.

**Changes in manuscript:** L23-26, P13, in the revised version: **"**Besides, the usability of total HCHO as the proxy of VOCs reactivity depends on time scale. Total HCHO can be regarded as the proxy for VOCs reactivity over a three-year study period, while only secondary HCHO can be used as the proxy for VOCs reactivity over a short study period, e.g. in winter."

7. So separating HCHO from different sources is full of great significance in sensitivity studies of O3 production.

**Responses:** Thanks. It was corrected.

**Changes in manuscript:** L26-27, P13, in the revised version: "So separating HCHO from different sources is of great significance insensitivity studies of $O_3$ production."

8. Line 12 in Page 2: "causing" should be "cause".

**Responses:** Thanks. It was corrected. Please see L12, P2 in the revised version.

9. Line 13 in Page 2: "favor to" should be "be in favor of".

**Responses:** Thanks. It was corrected. Please see L13, P2 in the revised version.

10. Line 21 in Page 5: "Surface air pollutants monitored by CNEMC was provided" should be "Surface air pollutants monitored by CNEMC were provided".

**Responses:** Thanks. It was corrected. Please see L25, P5 in the revised version.

11. Line 10 in Page 7: "the distribution of HCHO VCDs were homogenous "should be "the distribution of HCHO VCDs was homogenous ".

**Responses:** Thanks. It was corrected. Please see L18, P7 in the revised version.

12. Line 28 in Page 8: "The distribution of the three station selected to explore the primary and secondary sources of HCHO are shown" should be "The distribution of the three station selected to explore the primary and secondary sources of HCHO is shown".

**Responses:** Thanks. It was corrected. Please see L7, P9 in the revised version.

13. Line 4 in Page 11: The year "2015" should be "2005". The paper published in 2010 cannot research O3 pollution in 2015.

**Responses:** Thanks. It was corrected. Please see L16, P11 in the revised version.

14. Figure 3, the label of Yellow Sea and Bohai Sea goes against each other.

**Responses:** Thanks, we have fixed it in the revised version.

15. Figure 7: what are the P-value from the significance tests?

**Responses:** Thanks. The P-value from the significance tests were added in the Figure 7 captions.

---

## Author Comment (AC2) · 13 Mar 2019

We really appreciate your constructive comments and suggestions on our manuscript. Your positive advice help improve our manuscript. We have considered every comment carefully, and responded on a point to point and marked every change in red in the revised version.

**Specific comments:**

Page 6, section 3.2: the author estimated the total national cancer risk of 33500 people and 439 cancer cases per year using OMPS HCHO observation. Please explain how to convert?

**Responses:** Thank you very much for this suggestion. The total number of people who may develop cancer due to outdoor HCHO exposure in China is the sum of average cancer risk multiplied by the population in each provincial level administrative region. The total cancer cases per year are calculated through dividing the total number of people by life expectancy. And detailed formula and parameters are added in the revised version.

**Changes in manuscript:** L8-13, P7, in the revised version; L15, P7, in the revised version: "(Eq. (5))"

Page 6, Line 10-11: HCHO concentrations from OMPS measurement were generally lower than those from FTS, and the underestimation from OMPS was attributed to errors of spectral fitting and AMF calculation. The authors should explain the detailed reasons of errors instead of simply attributing spectral fitting and AMF calculation.

**Responses:** Thank you very much for this suggestion. Errors from spectral fitting affect the accuracy of SCD, and the uncertainties from the scattering weights calculated by the radiative transfer model and HCHO vertical profiles modeled by WRF-Chem affect AMF calculation. Spatial average from satellite observations and errors from SCD fitting and AMF calculation may causing the underestimation.

**Changes in manuscript:** L13-15, P6, in the revised version: "The underestimation from OMPS was attributed to spatial average from satellite observations and errors from HCHO SCDs affected by spectral fitting and AMF calculation affected by scattering weights calculated by the radiative transfer model and HCHO vertical profiles modeled by WRF-Chem."

Page 10, section 3.3.3 line 28-30: The authors conclude that primary emission influenced the variation of ambient HCHO more significantly than secondary formation in winter. This conclusion should be supported by quantitative analysis.

**Responses:** Thank you very much for this suggestion. At PDNA site, the correlation coefficient between total HCHO and primary HCHO (R=0.61) is larger than that between total HCHO and secondary HCHO (R=-0.42). So primary emission influenced more significantly than secondary formation on variation of ambient HCHO in winter.

**Changes in manuscript:** L7-9, P11, in the revised version: "While at PDNA site, primary emission (R=0.61, Table 4) influenced more significantly than secondary formation (R=-0.42, Table 4) on variation of ambient HCHO in winter and became the most important source to ambient HCHO in spring and winter of 2016."

Page 11, section 4.1: The author discussed whether total HCHO can be regarded as the proxy

for VOCs reactivity depending on the correlation between total HCHO with secondary HCHO. Why the relationship between them can be used to judge the representation of HCHO as the proxy for VOC reactivity?

**Responses:** Thank you very much for this suggestion. Secondary HCHO is an oxidation product of most VOCs and is produced with the formation of peroxy radicals ($RO_2$), positively correlating with $RO_2$ *(valin et al, 2015, ACP, wolfe et al, 2016, ACP)*. So only HCHO from secondary formation can be identified as the proxy for total VOCs reactivity. If total HCHO shows a good correlation with secondary HCHO, and then total HCHO will positively correlate with $RO_2$ and can be identified as the proxy for total VOCs reactivity.

**Changes in manuscript:** L22-23, P11, in the revised version: "The good correlation between total HCHO with secondary HCHO means a good correlation between total HCHO with $RO_2$, representing total HCHO as the proxy for VOCs reactivity."

Page 11, section 4.2: When discussing the HCHO control measures, it should be focused on HCHO pollution, i.e., HCHO concentrations beyond the air quality standard. When HCHO concentration is low, it is not necessary to discuss whether paying more attention to primary emission or secondary formation

**Responses:** Thank you very much for this suggestion. Here we define HCHO pollution with HCHO concentration reached the threshold value of 10 ppbv, the 95th percentile of all the observation data. HCHO pollution events occurred mostly in summer accounting for 81.0%, 81.3%, and 95.2% in Nanjing, Hangzhou and Shanghai. In summer, HCHO pollution occurred mostly during the periods from 24 to 27 July 2016 and 22 to 27 July 2017 in Nanjing, and from 23 to 27 July 2016 and from 20 to 27 July 2017 in Hangzhou. HCHO pollution in Shanghai lasts longer than Nanjing and Hangzhou, from 10 to 27 July of 2016 and 2017. Besides, HCHO pollution events were also observed during the period from 23 to 27 August 2015 and from 5 to 24 August 2017 in north of Shanghai center. In Shanghai suburb, three HCHO pollution periods were also observed, including: (1) from 19 June to 5 July 2015, (2) from 19 to 23 August 2016, (3) from 4 to 7 August 2017. During HCHO pollution events, the secondary formation contributed most to ambient HCHO and increased more largely than primary HCHO. Compared to the concentrations before pollution days, secondary HCHO and primary HCHO increased 0.90 ppbv and -0.12 ppbv in average, respectively, in Nanjing; increased 1.33 ppbv and 0.21 ppbv, respectively, in Hangzhou; and increased 1.26 ppbv and 0.14 ppbv, respectively, in Shanghai. So decreasing VOCs emissions, the source of secondary HCHO, has a more sensitive effect on controlling HCHO pollution in Nanjing, Hangzhou and Shanghai.

**Changes in manuscript:** L5-17, P12, in the revised version: "Here we define HCHO pollution with HCHO concentration reached the threshold value of 10 ppbv, the 95th percentile of all the observation data. HCHO pollution events occurred mostly in summer accounting for 81.0%, 81.3%, and 95.2% in Nanjing, Hangzhou and Shanghai. In summer, HCHO pollution occurred mostly during the periods from 24 to 27 July 2016 and 22 to 27 July 2017 in Nanjing, and from 23 to 27 July 2016 and from 20 to 27 July 2017 in Hangzhou. HCHO pollution in Shanghai lasts longer than Nanjing and Hangzhou, from 10 to 27 July of 2016 and 2017. Besides, HCHO pollution events were also observed during the period from 23 to 27 August 2015 and from 5 to 24 August 2017 in north of Shanghai center. In Shanghai

suburb, three HCHO pollution periods were also observed, including: (1) from 19 June to 5 July 2015, (2) from 19 to 23 August 2016, (3) from 4 to 7 August 2017. During HCHO pollution events, the secondary formation contributed most to ambient HCHO and increased more largely than primary HCHO. Compared to the concentrations before pollution days, secondary HCHO and primary HCHO increased 0.90 ppbv and -0.12 ppbv in average, respectively, in Nanjing; increased 1.33 ppbv and 0.21 ppbv, respectively, in Hangzhou; and increased 1.26 ppbv and 0.14 ppbv, respectively, in Shanghai. So decreasing VOCs emissions, the source of secondary HCHO, has a more sensitive effect on controlling HCHO pollution in Nanjing, Hangzhou and Shanghai."

**Technical corrections:**

Page2, Line4: change "become increasing serious" to "become increasingly serious"
**Responses:** Thanks. It was corrected. Please see L4, P2 in the revised version.

Page5, Line20: change "Surface air pollutants monitored by CNEMC was" to "Surface air pollutants monitored by CNEMC were"
**Responses:** Thanks. It was corrected. Please see L25, P5 in the revised version.

Page5, Line27: change "one of the condition" to "one of the conditions"
**Responses:** Thanks. It was corrected. Please see L1, P6 in the revised version.

Page8, Line11: change "industrial zoon" to "industrial zone"
**Responses:** Thanks. It was corrected. Please see L22, P8 in the revised version.

---

## Author Response (AR2)

We really appreciate you giving minor revisions on our manuscript, we have considered every comment carefully, and responded on a point to point and marked every change in red in the revised version.

**Comments:**

1. The MS addressed an important topic in viewing the variation of ambient HCHO by satellite. And the authors answered most of the comments raised by the two reviewers. I still have problem with the methods that the authors separating the primary and secondary sources of HCHO, by using simple linear regression. I double that secondary HCHO had different temporal variation with primary ones, so the linear regression of total HCHO would tend to correlated better with primary sources, but this did not mean less contribution of secondary sources, especially during photochemical reactive time period. I suggest that the authors seek a more reliable approaches to the apportion by more literature review in this regard.

**Responses:** Thank you very much for this suggestion. In our manuscript, multiple linear regression model was used to separate primary and secondary sources of ambient HCHO following Eq. (R1).

$$[C_{HCHO}]=\beta_0+\beta_1\times[C_{CO}]+\beta_2\times[C_{O_3}] \tag{R1}$$

where $\beta_0$, $\beta_1$, $\beta_2$ are the coefficients fitted by the model, $[C_{HCHO}]$, $[C_{CO}]$ and $[C_{O_3}]$ represent the concentrations or the transformations of concentrations for HCHO, CO, and $O_3$, respectively. In Garcia et al. (2006)'s study, the author applied ten different transformations: linear (no transformation), natural log, square root, second power, 3rd power, inverse, inverse of natural log, inverse of square root, inverse of second power and inverse of 3rd power on the time series of HCHO, CO, and $O_3$ following Eq. (R1). We transformed concentrations of HCHO, CO, and $O_3$ similar to Garcia et al. (2006)'s study at RJJ, CXT, and PDNA sites representing Nanjing, Hangzhou, and Shanghai respectively, and linear regression (no transformation) was identified as the best model (Table R1). The linear regression method was used widely in the source apportionment of HCHO observed by various methods, such as difference frequency generation (DFG) sensor (Friedfeld et al., 2002), fluorometric instrument (Li et al., 2010), proton transfer reaction ion-drift chemical ionization mass spectrometer (PTR-ID-CIMS) (Ma et al., 2016), 2,4-dinitrophenylhydrazine (DNPH) (Lui et al., 2017) and Multi Axis Differential Optical Absorption Spectroscopy (MAX-DOAS) (Hong et al., 2018). Here we apply it to deal with HCHO data from satellite observation. During the OMPS overpass time (13:30 LT), photochemical reaction is active. The results from linear regression model show that secondary HCHO contributed most to ambient HCHO and played a curial effect on variation of ambient HCHO in most time of study periods. Primary emission just dominated the variation of HCHO during some days in the winter.

**Changes in the manuscript:** L5-6, P8, in the revised version: "The statistical analysis of simultaneous real-time measurements of HCHO, CO, and $O_3$ can be described by a multiple regression model:"; L8-12, P8, in the revised version: "and $[C_{HCHO}]$, $[C_{CO}]$ and $[C_{O_3}]$ represent

the concentrations or the transformations of concentrations for HCHO, CO, and $O_3$, respectively. Similar to Garcia et al. (2006)'s study, we applied ten different transformations: linear (no transformation), natural log, square root, second power, 3rd power, inverse, inverse of natural log, inverse of square root, inverse of second power and inverse of 3rd power on the time series of HCHO, CO, and $O_3$, and linear regression without transformation was identified as the best model (Table S1).".

Table R1. Correlation coefficients of the regression analysis using different data transformations.

[revised manuscript text omitted]